# Time series experimental design under one-shot sampling: The importance of condition diversity

Xiaohan Kang[1]*, Bruce Hajek[1], Faqiang Wu[2], Yoshie Hanzawa[2]

**1** Coordinated Science Laboratory and Department of Electrical and Computer Engineering, University of Illinois at Urbana–Champaign, Urbana, Illinois, United States of America, **2** Department of Biology, California State University, Northridge, Northridge, California, United States of America

\* xiaohan.kang1@gmail.com

**Data Availability Statement:** The computer simulation code is available at https://github.com/Veggente/one-shot-sampling.

**Funding:** This work was supported by the Plant Genome Research Program from the National

## Abstract

Many biological data sets are prepared using one-shot sampling, in which each individual organism is sampled at most once. Time series therefore do not follow trajectories of individuals over time. However, samples collected at different times from individuals grown under the same conditions share the same perturbations of the biological processes, and hence behave as surrogates for multiple samples from a single individual at different times. This implies the importance of growing individuals under multiple conditions if one-shot sampling is used. This paper models the condition effect explicitly by using condition-dependent nominal mRNA production amounts for each gene, it quantifies the performance of network structure estimators both analytically and numerically, and it illustrates the difficulty in network reconstruction under one-shot sampling when the condition effect is absent. A case study of an *Arabidopsis* circadian clock network model is also included.

## Introduction

Time series data is important for studying biological processes in organisms because of the dynamic nature of the biological systems. Ideally it is desirable to use time series with *multi-shot sampling*, where each individual (such as a plant, animal, or microorganism) is sampled multiple times to produce the trajectory of the biological process, as in Fig 1. Then the natural biological variations in different individuals can be leveraged for statistical inference, and thus inference can be made even if the samples are prepared under the same experimental condition.

However, in many experiments multi-shot sampling is not possible. Due to stress response of the organisms and/or the large amount of cell tissue required for accurate measurements, the dynamics of the relevant biological process in an individual of the organism cannot be observed at multiple times without interference. For example, in an RNA-seq experiment an individual plant is often only sampled once in its entire life, leaving the dynamics unobserved at other times. See the Discussion section for a review of literature on this subject. We call the resulting time series data, as illustrated in Fig 2, a time series with *one-shot sampling*. Because

Science Foundation (NSF-IOS-PGRP-1823145) to B.H. and Y.H.

**Competing interests:** The authors have declared that no competing interests exist.

**Fig 1. Multi-shot sampling.** Each plant is observed four times.

the time series with one-shot sampling do not follow the trajectories of the same individuals, they do not capture all the correlations in the biological processes. For example, the trajectory of observations on plants 1–2–3–4 and that on 1–6–7–4 in Fig 2 are statistically identical. The resulting partial observation renders some common models for the biological system dynamics inaccurate or even irrelevant.

To address this problem, instead of getting multi-shot time series of single individuals, one can grow multiple individuals under each condition with a variety of conditions, and get one-shot time series of the single conditions. The one-shot samples from the same condition then become a surrogate for multi-shot samples for a single individual, as illustrated in Fig 3. In essence, if we view the preparation condition of each sample as being random, then there should be a positive correlation among samples grown under the same condition. We call this correlation the *condition variation effect*. It is similar to the effect of biological variation of a single individual sampled at different times, if such sampling were possible.

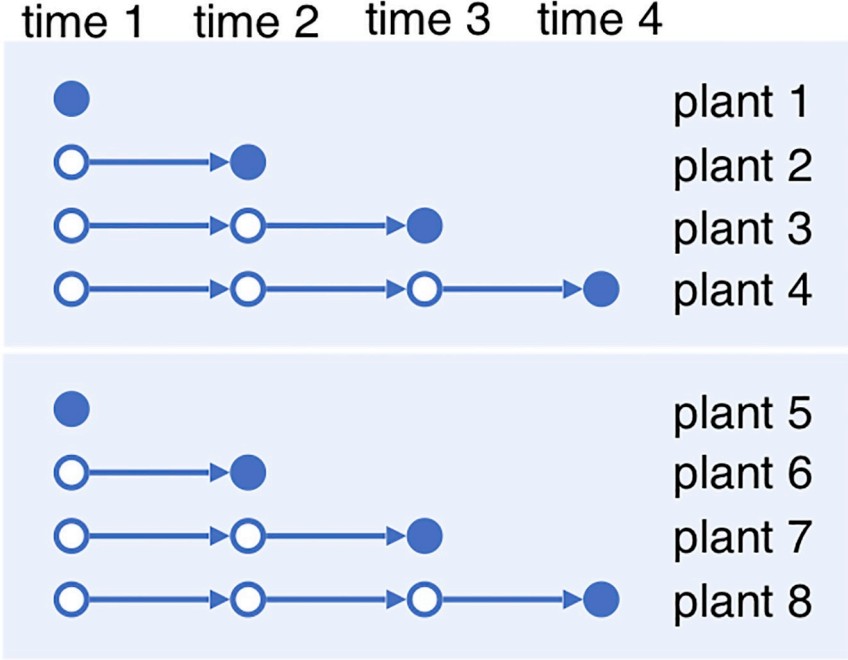

**Fig 2. One-shot sampling.** Each plant is observed once.

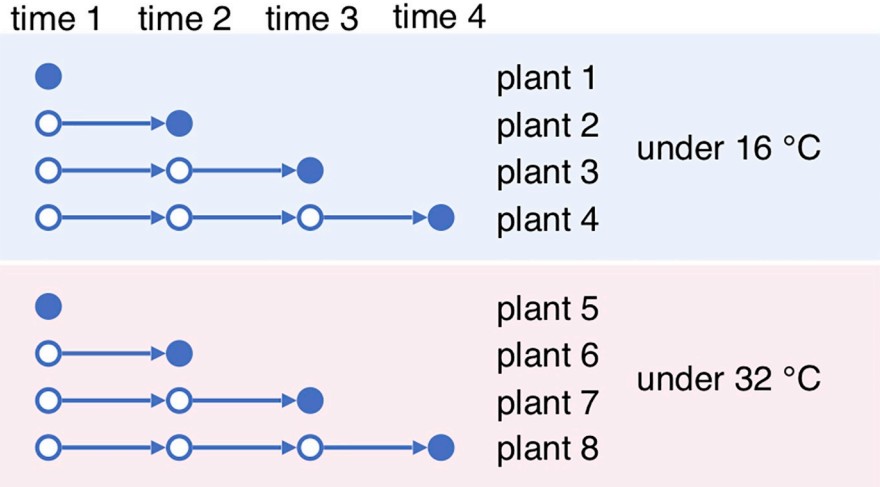

**Fig 3. One-shot sampling with two different conditions.**

For each condition, the one-shot samples at different times are also complemented by *biological replicates*, which are samples from independent individuals taken at the *same* time used to reduce technical and/or biological variations. See the Discussion section for a review on how replicates are used for biological inference. With a budget over the number of samples, a balance must be kept between the number of conditions, the number of sampling times and the number of replicates.

To illustrate and quantify the effect of one-shot sampling in biological inference, we introduce a simple dynamic gene expression model with a condition variation effect. We consider a hypothesis testing setting and model the dynamics of the gene expression levels at different sampling times by a dynamic Bayesian network (DBN), where the randomness comes from nominal (or basal) biological and condition variations for each gene. The nominal condition-dependent variation of gene $j$ is the same for all plants in that condition and the remaining variation is biological and is independent across the individuals in the condition. In contrast to GeneNetWeaver [1], where the effect of a condition is modeled by a random perturbation to the network coefficients, in our model the condition effect is characterized by correlation in the nominal variation terms of the dynamics. Note in both models samples from different individuals under the same condition are statistically independent given the randomness associated with the condition.

The contributions of this paper are threefold.

1. A composite hypothesis testing problem on the gene regulatory network is formulated and a gene expression dynamic model that explicitly captures the per-gene condition effect and the gene regulatory interactions is proposed.

2. The performance of gene regulatory network structure estimators is analyzed for both multi-shot and one-shot sampling, with focus on two algorithms. Furthermore, single-gene and multi-gene simulation results indicate that multiple-condition experiments can somewhat mitigate the shortcomings of one-shot sampling.

3. The difficulty of network reconstruction under one-shot sampling with no condition effect is illustrated. This difficulty connects network analysis and differential expression analysis,

two common tasks in large-scale genomics studies, in the sense that the part of network involving non-differentially expressed genes may be harder to reconstruct.

The simulation code for generating the figures is available at [2].

# Materials and methods

## Stochastic model of time-series samples

This section formulates the hypothesis testing problem of learning the structure of the gene regulatory network (GRN) from gene expression data with one-shot or multi-shot sampling. The GRN is characterized by an unknown adjacency matrix. Given the GRN, a dynamic Bayesian network model is used for the gene expression evolution with time. Two parameters $\sigma_{\mathrm{co},j}$ and $\sigma_{\mathrm{bi},j}$ are used for each gene $j$, with the former explicitly capturing the condition variation effect and the latter capturing the biological variation level.

**Notation.** For any positive integer $n$, let $[n] = \{1, 2, \ldots, n\}$. We use $(f(x))_{x \in \mathcal{I}}$ to denote the family of elements in the set $\{f(x) : x \in \mathcal{I}\}$ indexed by the index set $\mathcal{I}$. The indicator function on a statement or a set $P$ is denoted by $\mathbb{1}_P$. The $n$-by-$n$ identity matrix is denoted by $I_n$. The transpose of matrix $A$ is denoted by $A^*$.

**Model for gene regulatory network topology.** Let $n$ be the number of genes and let $A \in \mathcal{A} \subseteq \mathbb{R}^{n \times n}$ be the unknown adjacency matrix of the GRN. The sign of the entry $a_{ij}$ of $A$ for $i \neq j$ indicates the type of regulation of $j$ by $i$, and the absolute value the strength of the regulation. A zero entry $a_{ij} = 0$ with $i \neq j$ indicates no regulation of $j$ by $i$. The diagonal of $A$ characterizes protein concentration passed from the previous time, protein degradation, and gene autoregulation. Let $\mathcal{S} = \{S_1, S_2, \ldots, S_{|\mathcal{S}|}\}$ be a finite set of network structures and let $D$ be a mapping from $\mathcal{A}$ to $\mathcal{S}$; $D(A)$ represents the network structure of an adjacency matrix $A$. Then $\mathcal{A}$ is partitioned by the associated network structures. Fix a loss function $l : \mathcal{S}^2 \to \mathbb{R}$. Let $Y \in \mathcal{Y}$ be the random observation and let $\delta : \mathcal{Y} \to \mathcal{S}$ be an estimator for the structure. The performance of an estimator is evaluated by the expected loss $\mathbb{E}l(D(A), \delta(Y))$. This is a hypothesis testing problem with composite hypotheses $\{D^{-1}(S) : S \in \mathcal{S}\}$. This paper considers network reconstruction up to regulation type with $D(A) = (\mathrm{sgn}(A_{ij}))_{(i,j) \in [n]^2}$, where $\mathrm{sgn}(s) = \mathbb{1}_{\{s > 0\}} - \mathbb{1}_{\{s < 0\}}$. In other words, the ternary value of the edge signs (positive, negative, or no edge) are to be recovered. A structure $S$ has the form $S = (S_{ij})_{(i,j) \in [n]^2}$ with $S_{ij} \in \{0, 1, -1\}$, and it can be interpreted as a directed graph with possible self-loops. Some examples of loss functions are as follows.

- Ternary false discovery rate (FDR)

$$l_{\mathrm{FDR}}(S, S') = 1 - \frac{\sum_{i=1}^n \sum_{j=1}^n \mathbb{1}_{\{S_{ij} = S'_{ij} \neq 0\}}}{\sum_{i=1}^n \sum_{j=1}^n \mathbb{1}_{\{S'_{ij} \neq 0\}}}.$$

- Ternary false negative rate (FNR)

$$l_{\mathrm{FNR}}(S, S') = 1 - \frac{\sum_{i=1}^n \sum_{j=1}^n \mathbb{1}_{\{S_{ij} = S'_{ij} \neq 0\}}}{\sum_{i=1}^n \sum_{j=1}^n \mathbb{1}_{\{S_{ij} \neq 0\}}}.$$

- Ternary false positive rate (FPR)

$$l_{\text{FPR}}(S, S') = 1 - \frac{\sum_{i=1}^{n} \sum_{j=1}^{n} \mathbb{1}_{\{S_{ij} = S'_{ij} = 0\}}}{\sum_{i=1}^{n} \sum_{j=1}^{n} \mathbb{1}_{\{S_{ij} = 0\}}}.$$

- Ternary error rate

$$l_{\text{E}}(S, S') = \frac{1}{n^2} \sum_{i=1}^{n} \sum_{j=1}^{n} \mathbb{1}_{\{S_{ij} \neq S'_{ij}\}}.$$

Note the FDR and the FNR are well-defined when $S'$ and $S$ contains nonzero elements, respectively, and the FPR is well-defined when $S$ contains zeros. The error rate is always well-defined. It can be seen that $l_{\text{FDR}}(S, S') = l_{\text{FNR}}(S', S)$. Also if $S$ does not contain zeros then $l_{\text{FNR}}(S, S') = l_{\text{E}}(S, S')$. Similarly if $S'$ does not contain zeros then $l_{\text{FNR}}(S, S') = l_{\text{E}}(S, S')$. As an example, for a random guessing algorithm with probabilities of $S'_{ij} = 0, 1, -1$ being $1 - q, q/2, q/2$ and a network prior with probabilities of $S_{ij} = 0, 1, -1$ being $1 - p, p/2, p/2$, $l_{\text{FDR}} = 1 - p/2$, $l_{\text{FNR}} = 1 - q/2$, and $l_{\text{FPR}} = q$.

**Model for gene expression dynamics.** This section models the gene expression dynamics of individuals by a dynamic Bayesian networks with parameters $\sigma_{\text{co},j}$ and $\sigma_{\text{bi},j}$ as the condition variation level and biological variation level for gene $j$.

Let $K$, $T$ and $C$ be the number of individuals, sampling times, and conditions, respectively. Let $X_j^k(t) \in \mathbb{R}$ be the expression level of gene $j \in [n]$ in individual $k \in [K]$ at time $t \in [T]$, and let $c_k \in [C]$ be the label that indicates the condition for individual $k$. Here we assume $X_j^k(t)$ represents both the mRNA abundance and the protein concentration. The gene expression levels evolve according to the *Gaussian linear model* (GLM) with initial condition $X_j^k(0) = 0$ for any $j \in [n]$, $k \in [K]$ and the following recursion (note the values of $X$ can be the expression levels after a logarithm transform, in which case lowly expressed genes have negative $X$ values)

$$X_j^k(t + 1) = \sum_{i=1}^{n} X_i^k(t) A_{ij} + \sigma_{\text{co},j} W_{\text{co},j}^{c_k}(t + 1) + \sigma_{\text{bi},j} W_{\text{bi},j}^k(t + 1), \tag{1}$$

for $j \in [n]$, $k \in [K]$, and $t \in \{0, 1, \ldots, T-1\}$, where $\left(W_{\text{co},j}^c(t)\right)_{(c,j,t) \in [C] \times [n] \times [T]}$ and $\left(W_{\text{bi},k}^j(t)\right)_{(j,k,t) \in [n] \times [K] \times [T]}$ are collections of independent standard Gaussian random variables that are used to drive the dynamics. Here the last two terms in (1) denote the condition variation and biological variation, respectively. To write (1) in matrix form, we let $X(t) = \left(X_j^k(t)\right)_{(k,j) \in [K] \times [n]}$ and $W(t) = \left(W_j^k(t)\right)_{(k,j) \in [K] \times [n]}$ be $K$-by-$n$ matrices, where $W_j^k(t) = \sigma_{\text{co},j} W_{\text{co},j}^{c_k}(t) + \sigma_{\text{bi},j} W_{\text{bi},j}^k(t)$. Then

$$X(t + 1) = X(t)A + W(t + 1) \tag{2}$$

and hence

$$X(t) = \sum_{\tau=1}^{t} W(\tau) A^{t-\tau}. \tag{3}$$

The variable $W_j^k(t)$ is the nominal mRNA production amount for target gene $j$, individual $k$ at time $t$ that would occur in the absence of regulation by other genes.

**Model for sampling method.**   In this section two sampling methods are described: one-shot sampling and multi-shot sampling. For simplicity, throughout this paper we consider a *full factorial design* with $CRT$ samples obtained under $C$ conditions, $R$ replicates and $T$ sampling times, denoted by $Y = (Y^{c,r,t})_{(c,r,t) \in [C] \times [R] \times [T]}$. In other words, instead of $X$ we observe $Y$, a noisy and possibly partial observation of $X$. Here the triple index for each sample indicates the condition, replicate, and time. As we will see in the Discussion at the end of this section, for either sampling method, the biological variation level $\sigma_{\mathrm{bi},j}$ can be reduced by combining multiple individuals to form a single sample.

**Multi-shot sampling.**   Assume an individual can be sampled multiple times. This sampling model corresponds to $K = CR$ and $c_k = \lceil \frac{k}{R} \rceil \in [C]$ for all $k \in [K]$. Equivalently, multi-index $(c, r)$ can be used to determine the individual instead of $k$ for $X$ and $W$ with $c$ denoting the condition and $r$ the replicate. Then (1) for multi-shot sampling can be rewritten as

$$X_j^{c,r}(t+1) = \sum_{i=1}^{n} X_i^{c,r}(t) A_{ij} + \sigma_{\mathrm{co},j} W_{\mathrm{co},j}^c(t+1) + \sigma_{\mathrm{bi},j} W_{\mathrm{bi},j}^{c,r}(t+1),$$

and the observation for condition $c$, replicate $r$ and time $t$ is

$$Y_j^{c,r,t} = X_j^{c,r}(t) + \sigma_{\mathrm{te},j} Z_j^{c,r,t}, \tag{4}$$

with $(Z_j^{c,r,t})_{(j,c,r,t) \in [n] \times [C] \times [R] \times [T]}$ being a collection of independent standard Gaussian random variables modeling the observation noise, and $\sigma_{\mathrm{te},j}$ is the technical variance level of gene $j$. We see that for fixed $c$ and $r$ the observations at different times are from the same individual with the multi-index $(c, r)$. As a result, with multi-shot sampling $Y$ is a noisy full observation of $X$.

**One-shot sampling.**   Assume an individual can be sampled only once. This model corresponds to $K = CRT$ and $c_k = \lceil \frac{k}{RT} \rceil \in [C]$ for all $k \in [K]$. Equivalently, with multi-index $(c, r, s)$ denoting the condition, the replicate, and the target sampling time, the evolution (1) for one-shot sampling can be rewritten as

$$X_j^{c,r,s}(t+1) = \sum_{i=1}^{n} X_i^{c,r,s}(t) A_{ij} + \sigma_{\mathrm{co},j} W_{\mathrm{co},j}^c(t+1) + \sigma_{\mathrm{bi},j} W_{\mathrm{bi},j}^{c,r,s}(t+1),$$

and the observation is

$$Y_j^{c,r,t} = X_j^{c,r,t}(t) + \sigma_{\mathrm{te},j} Z_j^{c,r,t}. \tag{5}$$

Again $\sigma_{\mathrm{te},j}$ is the observation noise level of gene $j$ and the $Z$'s are independent standard Gaussian random variables. Note that for fixed $c$ and $r$ the observations at different times are from *different* individuals because the triple indices are different. Hence with one-shot sampling, $Y$ is a noisy *partial* observation of $X$ (to see this, note for gene 1 and the individual indexed by condition 1, replicate 1, and target sampling time 1, $X_1^{1,1,1}(1)$, which is the expression level at time 1, is observed through $Y_1^{1,1,1}$ but $X_1^{1,1,1}(2)$, which is the expression level at time 2, is not observed).

**Discussion on sources of variance.**   The $\sigma_{\mathrm{co},j} W_{\mathrm{co},j}^c(t)$ terms measure the condition-dependent nominal production level as global driving noise terms that are shared across individuals under the same condition. They are independent and identically distributed (i.i.d.) across conditions. The $\sigma_{\mathrm{bi},j} W_{\mathrm{bi},j}^k(t)$ terms measure the biological nominal production level of individuals as local driving noise terms. They are i.i.d. across individuals. The $\sigma_{\mathrm{te},j} Z_j^{c,r,t}$ terms measure the technical variation of samples as additive observational noise terms that are not in the evolution of $X$. They are i.i.d. across samples. We then have the following observations.

1. If the samples of the individuals under many different conditions are averaged and treated as a single sample, then effectively $\sigma_{\mathrm{co},j}$, $\sigma_{\mathrm{bi},j}$ and $\sigma_{\mathrm{te},j}$ are reduced.

2. If the samples of $R$ individuals under same conditions (biological replicates) are averaged and treated as a single sample, then effectively $\sigma_{\mathrm{bi},j}^2$ and $\sigma_{\mathrm{te},j}^2$ are reduced by a factor of $R$ while $\sigma_{\mathrm{co},j}^2$ remains unchanged.

3. If material from multiple individuals grown under the same condition is combined into a composite sample before measuring, then effectively $\sigma_{\mathrm{bi},j}$ is reduced while $\sigma_{\mathrm{co},j}$ and $\sigma_{\mathrm{te},j}$ remain unchanged. Note for microorganisms a sample may consist of millions of individuals and the biological variation is practically eliminated ($\sigma_{\mathrm{bi},j} \approx 0$).

4. If the samples from same individuals (technical replicates) are averaged and treated as a single sample, then effectively $\sigma_{\mathrm{te},j}$ is reduced while $\sigma_{\mathrm{co},j}$ and $\sigma_{\mathrm{bi},j}$ remain unchanged.

Note this sampling model with hierarchical driving and observational noises can also be used for single-cell RNA sequencing (scRNAseq) in addition to bulk RNA sequencing and microarray experiments. For scRNAseq, $\sigma_{\mathrm{co},j}$ can model the tissue-dependent variation (the global effect) and $\sigma_{\mathrm{bi},j}$ the per-cell variation (the local effect).

## Results

### Performance evaluation of network structure estimators

This section studies the performance of network structure estimators with multi-shot and one-shot sampling data. First, general properties of the two sampling methods are obtained. Then two algorithms, the generalized likelihood-ratio test (GLRT) and the basic sparse linear regression (BSLR), are studied. The former is a standard decision rule for composite hypothesis testing problems and is shown to have some properties but is computationally infeasible for even a small number of genes. The latter is an algorithm based on linear regression, and is feasible for a moderate number of genes. Finally simulation results for a single-gene network with GLRT and for a multi-gene network with BSLR are shown.

**General properties.** By (3), (4) and (5), the samples $Y$ are jointly Gaussian with zero mean. The covariance of the random tensor $Y$ is derived under the two sampling methods in the following.

Under multi-shot sampling, the samples under different conditions are independent and hence uncorrelated. Consider $Y^{c,r,t}$ and $Y^{c,r',t'}$, which are two samples under the same condition and collected at times $t$ and $t'$. The covariance matrix between $Y^{c,r,t}$ and $Y^{c,r',t'}$ is the sum of the covariance matrices of their common variations at times $\tau$ for $1 \le \tau \le t \wedge t'$ multiplied by $(A^*)^{t-\tau}$ on the left and $A^{t'-\tau}$ on the right, plus covariance for the observation noise. Let $\Sigma_{\mathrm{co}} = \mathrm{diag}(\sigma_{\mathrm{co},1}^2, \sigma_{\mathrm{co},2}^2, \ldots, \sigma_{\mathrm{co},n}^2)$, $\Sigma_{\mathrm{bi}} = \mathrm{diag}(\sigma_{\mathrm{bi},1}^2, \sigma_{\mathrm{bi},2}^2, \ldots, \sigma_{\mathrm{bi},n}^2)$, and $\Sigma_{\mathrm{te}} = \mathrm{diag}(\sigma_{\mathrm{te},1}^2, \sigma_{\mathrm{te},2}^2, \ldots, \sigma_{\mathrm{te},n}^2)$. Then the covariance matrix of the variations is $\Sigma_{\mathrm{co}} + \Sigma_{\mathrm{bi}}$ if the two samples are from the same individual (i.e., $r = r'$), and $\Sigma_{\mathrm{co}}$ otherwise. This yields:

$$
\begin{aligned}
&\mathbb{E}\left[(Y^{c,r,t})^* Y^{c',r',t'}\right] \\
&= \begin{cases}
\sum_{\tau=1}^{t} (A^*)^{t-\tau}(\Sigma_{\mathrm{co}} + \Sigma_{\mathrm{bi}})A^{t-\tau} + \Sigma_{\mathrm{te}} & \text{if } (c,r,t) = (c',r',t'), \\
\sum_{\tau=1}^{t\wedge t'} (A^*)^{t-\tau}(\Sigma_{\mathrm{co}} + \Sigma_{\mathrm{bi}})A^{t'-\tau} & \text{if } (c,r) = (c',r') \text{ and } t \ne t', \\
\sum_{\tau=1}^{t\wedge t'} (A^*)^{t-\tau}\Sigma_{\mathrm{co}}A^{t'-\tau} & \text{if } c = c' \text{ and } r \ne r', \\
0 & \text{if } c \ne c'.
\end{cases}
\end{aligned}
$$

Under one-shot sampling the only difference compared with multi-shot sampling is that two samples indexed by $(c, r, t)$ and $(c, r, t')$ are from different individuals if $t \neq t'$. So

$$
\mathbb{E}[(Y^{c,r,t})^* Y^{c',r',t'}]
$$
$$
=
\begin{cases}
\sum_{\tau=1}^{t} (A^*)^{t-\tau} (\Sigma_{\text{co}} + \Sigma_{\text{bi}}) A^{t-\tau} + \Sigma_{\text{te}} & \text{if } (c, r, t) = (c', r', t'), \\
\sum_{\tau=1}^{t \wedge t'} (A^*)^{t-\tau} \Sigma_{\text{co}} A^{t'-\tau} & \text{if } c = c' \text{ and } (r, t) \neq (r', t'), \\
0 & \text{if } c \neq c'.
\end{cases}
\tag{6}
$$

For any fixed network structure estimator:

1. If $\Sigma_{\text{bi}} = 0$ and $C$, $R$ and $T$ are fixed, the joint distribution of the data is the same for both types of sampling. So the performance of the estimator would be the same for multi-shot and one-shot sampling.

2. If $\Sigma_{\text{bi}} = 0$ and $\Sigma_{\text{te}} = 0$ (no observation noise) and $C$, $T$ are fixed, the joint distribution of the data is the same for both types of sampling (as noted in item 1) and any replicates beyond the first are identical to the first. So the performance of the estimator can be obtained even if all replicates beyond the first are discarded.

3. Under multi-shot sampling, when $C$, $R$, $T$ are fixed with $R = 1$, the joint distribution of the data depends on $\Sigma_{\text{co}}$ and $\Sigma_{\text{bi}}$ only through their sum. So the performance of the estimator would be the same for all $\Sigma_{\text{co}}$ and $\Sigma_{\text{bi}}$ such that $\Sigma_{\text{co}} + \Sigma_{\text{bi}}$ is the same.

4. In the homogeneous gene case with $\sigma_{\text{co},j} = \sigma_{\text{co}}$, $\sigma_{\text{bi},j} = \sigma_{\text{bi}}$, $\sigma_{\text{te},j} = \sigma_{\text{te}}$ for all $j$ with $\sigma_{\text{co}}^* + \sigma_{\text{bi}}^* + \sigma_{\text{te}}^* > 0$, suppose that the estimator $\delta$ is based on replicate averages $y = (y^{c,t})_{(c,t) \in [C] \times [T]}$ with $y^{c,t} = \frac{1}{R} \sum_{r=1}^{R} Y^{c,r,t}$, and that $\delta$ is scale-invariant (i.e., $\delta(Y) = \delta(c_0 Y)$ for any $c_0 \neq 0$ and $Y$). Then under multi-shot sampling, $\delta$'s performance depends on $\sigma_{\text{co}}$, $\sigma_{\text{bi}}$, $\sigma_{\text{te}}$ and $R$ only through the ratio $\frac{\sigma_{\text{te}}^2/R}{\sigma_{\text{co}}^2 + \sigma_{\text{bi}}^2/R + \sigma_{\text{te}}^2/R}$. Under one-shot sampling, the estimator's performance depends on $\sigma_{\text{co}}$, $\sigma_{\text{bi}}$, $\sigma_{\text{te}}$ and $R$ only through the ratios $\frac{\sigma_{\text{te}}^2/R}{\sigma_{\text{co}}^2 + \sigma_{\text{bi}}^2/R + \sigma_{\text{te}}^2/R}$ and $\frac{\sigma_{\text{co}}^2}{\sigma_{\text{co}}^2 + \sigma_{\text{bi}}^2/R}$ (through the latter only when $\sigma_{\text{co}}^2 + \sigma_{\text{bi}}^2 > 0$).

To see 4), recall from observation 2 above that averaging reduces the variance of the biological variation and that of the observation noise by a factor of $R$ due to independence, but preserves the condition variation because it is identical across replicates. Hence the variance of the driving noise in the averages is $\sigma_{\text{co}}^2 + \sigma_{\text{bi}}^2/R$ and the variance of the observation noise of the averages is $\sigma_{\text{te}}^2/R$. Then the averages are essentially single-replicate data, and the performance under multi-shot sampling depends only on the ratio of the new driving noise variance to the new observational noise variance. For one-shot sampling the ratio between the condition variation and the biological variation also matters for the single-replicate data when the condition variation and the biological variation are not both zero, so the performance also depends on $\frac{\sigma_{\text{co}}^2}{\sigma_{\text{co}}^2 + \sigma_{\text{bi}}^2/R}$.

**Network reconstruction algorithms.** In this section we introduce GLRT and BSLR. GLRT is a standard choice in composite hypothesis testing setting. We observe some properties for GLRT under one-shot and multi-shot sampling. However, GLRT involves optimizing the likelihood over the entire parameter space, which grows exponentially with the square of the number of genes. Hence GLRT is hard to compute for multiple-gene network reconstruction. In contrast, BSLR is an intuitive algorithm based on linear regression, and will be shown in simulations to perform reasonably well for multi-gene scenarios.

**GLRT.** The GLRT (see, e.g., page 38, Chapter II.E in [3]) is given by $\delta(y) = D(\hat{A}_{\text{ML}}(y))$, where $\hat{A}_{\text{ML}}(y)$ is the maximum-likelihood estimate for $A$ based on the covariance of $Y$ given the observation $Y = y$.

**Proposition 1** *GLRT (with the knowledge of $\Sigma_{\text{co}}$, $\Sigma_{\text{bi}}$ and $\Sigma_{\text{te}}$) has the following properties.*

1. *For a fixed $\sigma^2$, under multi-shot sampling with $\Sigma_{\text{te}} = 0$ (no observation noise), $\sigma_{\text{co},j} = \sigma_{\text{co}}$, $\sigma_{\text{bi},j} = \sigma_{\text{bi}}$, and $\sigma_{\text{co}}^2 + \sigma_{\text{bi}}^2 = \sigma^2$, the performance of GLRT for sign estimation is the same for all $(R, \sigma_{\text{co}}, \sigma_{\text{bi}})$ excluding $(R \geq 2, \sigma_{\text{bi}} = 0)$.*

2. *Under one-shot sampling and $\Sigma_{\text{co}} = 0$, the log likelihood of the data as a function of $A$ (i.e. the log likelihood function) is invariant with respect to replacing $A$ by $-A$. So, for the single-gene $n = 1$ case, the log likelihood function is an even function of $A$, and thus the GLRT will do no better than random guessing.*

For 2 it suffices to notice in (6) the covariance is invariant with respect to changing $A$ to $-A$. A proof of 1 is given below.

*Proof* of 1). We first prove it for the case of a single gene with constant $T$ and a constant number of individuals, $CR$. To do that we need to look at the likelihood function closely.

We may assume $\sigma^2 = 1$. Because the trajectories for different conditions are independent (for given parameters $(A, \sigma_{\text{co}}^2)$), we shall first consider the case with a single condition; i.e., $C = 1$. There are hence $R$ trajectories of length $T$. Then the covariance matrix of the length-$R$ driving vector used at time $t$ for the trajectories is

$$\text{Cov}(W(t)) = (1 - \sigma_{\text{co}}^2)I_R + \sigma_{\text{co}}^2 J_R =: \Sigma.$$

When $\sigma_{\text{co}} > 0$, $\Sigma$ is not the identity matrix multiplied by some constant; i.e., the noise vector $W(t)$ is colored across replicates. It can be checked when $\sigma_{\text{co}} < 1$ (i.e., $\sigma_{\text{bi}} > 0$) the matrix $\Sigma$ is positive definite. Then there exists an orthogonal matrix $U$ and a diagonal matrix $\Lambda$ with positive diagonal elements such that $\Sigma = U\Lambda U^*$. Let $\Sigma^{-1/2} = U\Lambda^{-1/2} U^*$ and let

$$\tilde{X}(t) = \Sigma^{-1/2}X(t),$$
$$\tilde{W}(t) = \Sigma^{-1/2}W(t)$$

for all $t \in [T]$. Then the trajectories for the $R$ replicates in a single condition become:

$$\tilde{X}(t + 1) = \tilde{X}(t)A + \tilde{W}(t + 1).$$

It can be checked that after the linear transformation by $\Sigma^{-1/2}$, which does not depend on $A$, the new driving vectors are white (i.e., $\text{Cov}(\tilde{W}(t)) = I_R$). It follows that the distribution of $\tilde{X}|(A, \sigma_{\text{co}}^2)$ is the same as the distribution of $X|(A, 0)$ (i.e. $\sigma_{\text{co}} = 0$). Therefore, for $x = (x^r(t))_{(r,t) \in [R] \times [T]}$, if we let $L_X(x|A, \sigma_{\text{co}}^2)$ denote the likelihood of $X = x$ for parameters $A, \sigma_{\text{co}}^2$, then

$$L_X(x|A, \sigma_{\text{co}}^2) = L\tilde{X}(\tilde{x}|A, \sigma_{\text{co}}^2)d(R, T, \sigma_{\text{co}}^2) = L_X(\tilde{x}|A, 0)d(R, T, \sigma_{\text{co}}^2),$$

where $d(R, T, \sigma_{\text{co}}^2) = (\det \Sigma)^{-T/2}$ is a function of $R$, $T$ and $\sigma_{\text{co}}^2$, and $\tilde{x}(t) = \Sigma^{-1/2}x(t)$.

Now consider the likelihood function for all $CRT$ samples with general $C$. It is the product of $C$ likelihood functions for the samples prepared under the $C$ different conditions. It is thus equal to $d(R, T, \sigma_{\text{co}}^2)^C$ times the likelihood of the transformed expression levels $\tilde{x}$, which is the likelihood function for $\sigma_{\text{co}} = 0$ and a total of $CRT$ samples. The form of the product depends on $C$ and $R$ only through $CR$, because under the transformation, all $CR$ trajectories are independent. Hence, for fixed $A, \sigma_{\text{co}}^2, C, R, T$ the distribution of the maximum likelihood estimate

of $A$, when the samples are generated using a given $\sigma_{co} > 0$ (so the $R$ individuals under each condition are correlated) and the likelihood function also uses $\sigma_{co}^2$, is the same as the distribution of the maximum likelihood estimate of $A$ when $\sigma_{co} = 0$ (in which case the $CR$ individual trajectories are i.i.d.). Formally,

$$
\begin{aligned}
\mathbb{E}_{\sigma_{co}} l(D(A), \delta(Y)) \quad &= \mathbb{E}_{\sigma_{co}} l(D(A), D(\arg \max_{A'} L_X(X|A', \sigma_{co}^2))) \\
&= \mathbb{E}_{\sigma_{co}} l(D(A), D(\arg \max_{A'} L_X(\tilde{X}|A', 0))) \\
&= \mathbb{E}_0 l(D(A), D(\arg \max_{A'} L_X(X|A', 0))) \\
&= \mathbb{E}_0 l(D(A), \delta(Y)),
\end{aligned}
$$

where $\mathbb{E}_{\sigma_{co}}$ denotes that the condition variation level of the random elements $X$ and $Y$ is $\sigma_{co}^2$. The above fails if $\sigma_{co} = 1$ (i.e., $\sigma_{bi} = 0$) and $R \geq 2$ because then $\Sigma$ is singular. It also fails if $\sigma_{co}$ and $\sigma_{bi}$ are unknown to the GLRT.

For the general model with multiple genes, if $\sigma_{co,j}$ is the same for each gene $j$, 1) holds as before—for the proof, apply left multiplication by $\Sigma^{-\frac{1}{2}}$ for each gene, time, and condition to all $R$ samples in the condition. View (2) as an update equation for an $R \times n$ matrix for each group of $R$ samples in one condition. One column of length $R$ per gene, and one row per sample.

**BSLR.** In BSLR, replicates are averaged and the average gene expression levels at different times under different conditions are fitted in a linear regression model with best-subset sparse model selection, followed by a Granger causality test to eliminate the false discoveries. BSLR is similar to other two-stage linear regression–based network reconstruction algorithms, notably oCSE [4] and CaSPIAN [5]. Both oCSE and CaSPIAN use greedy algorithms in the first build-up stage, making them more suitable for large-scale problems. In contrast, BSLR uses best subset selection, which is conceptually simpler but computationally expensive for large $n$. For the tear-down stage both BSLR and CaSPIAN use the Granger causality test, while oCSE uses a permutation test.

**Build-up stage.** In the first stage BSLR finds potential regulatory interactions using a linear regression model. Let $Y_j^c(t) = \frac{1}{R} \sum_{r=1}^R Y^{c,r,t}$ and let $Y(t) = (Y_j^c(t))_{(c,j) \in [C] \times [n]}$ denote the $C$-by-$n$ matrix. Let

$$
\Psi(1) = \begin{pmatrix} Y(2) \\ Y(3) \\ \vdots \\ Y(T) \end{pmatrix}
$$

and

$$
\Psi(0) = \begin{pmatrix} Y(1) \\ Y(2) \\ \vdots \\ Y(T-1) \end{pmatrix}.
$$

For each target gene $j \in [n]$, BSLR solves the following best subset selection problem with a

subset size $k < n$:

$$\underset{A_{\cdot j}, b_j, d_j}{\text{minimize}} \quad \| \Psi_j(1) - \Psi(0)A_{\cdot j} - d_j\Psi_j(0) - b_j\mathbf{1} \|_2^2$$

$$\text{subject to} \quad \|A_{\cdot j}\|_0 \leq k \text{ and } A_{jj} = 0.$$

Denote the solution by $(A^*, b^*, d^*)$. The output of the first stage is then $A^*$.

A naive algorithm to solve the above optimization has a computational complexity of $O(n^{k+1})$ for fixed $k$ as $n \to \infty$. Faster near-optimal alternatives exist [6].

**Tear-down stage.** The second stage is the same as that of CaSPIAN. For each $j \in [n]$ and each $i \in \text{supp}(A^*_{\cdot j})$, let the unrestricted residual sum of squares be

$$\text{RSS}_u = \| \Psi_j(1) - \Psi(0)A^*_{\cdot j} - d^*_j\Psi_j(0) - b^*_j\mathbf{1} \|_2^2$$

and the restricted residual sum of squares

$$\text{RSS}_r = \inf\{\| \Psi_j(1) - \Psi(0)A_{\cdot j} - d_j\Psi_j(0) - b_j\mathbf{1} \|_2^2:$$
$$\text{supp}(A_{\cdot j}) = \text{supp}(A^*_{\cdot j})\backslash\{i\}\}.$$

The *F*-statistic is given by

$$F = \frac{\text{RSS}_r - \text{RSS}_u}{\text{RSS}_u/(C(T-1)-k-2)}.$$

The potential parent $i$ of $j$ is removed in the tear-down stage if the $p$-value of the $F$-statistic with degrees of freedom $(1, C(T-1)-k-2)$ is above the preset significance level (e.g., 0.05). Note the tests are done for all parents in $A_{\cdot j}$ simultaneously; both the restricted and the unrestricted models contain the other potential parents regardless of the results of the tests on them.

**Simulations on single-gene network reconstruction.** The GLM is used to simulate one-shot sampling data with a single gene. The goal is to determine the type of autoregulation of the single gene (activation or repression). The protein concentration passed from the previous time is ignored so the type of autoregulation is represented by the sign of the scalar $A$. In order to compare one-shot and multi-shot sampling, we view the main expense to be proportional to the number of samples to prepare as opposed to the number of individuals to grow. We thus fix a total budget of $CRT = 180$ samples and consider full factorial design with $C$ and $R$ varying with $CR = 30$, and $T = 6$ with 10 000 simulations. We assume the knowledge of the existence of the autoregulation (i.e., $A \neq 0$), in which case the FDR, the FNR and the error rate coincide, so we only look at error rates. The results are plotted in Fig 4. The four plots on the left are for one-shot sampling and the four on the right are for multi-shot sampling. Consider the homogeneous case with $\sigma_{\text{co},j} = \sigma_{\text{co}}$, $\sigma_{\text{bi},j} = \sigma_{\text{bi}}$ and $\sigma_{\text{te},j} = \sigma_{\text{te}}$ for all $j$ and let $\gamma = \frac{\sigma_{\text{co}}^2}{\sigma_{\text{co}}^2 + \sigma_{\text{bi}}^2}$ be the fraction of condition variation in the driving noise. For each plot the observed probability of sign (of $A$) error is shown for $\gamma \in \{0, 0.2, 0.4, 0.6, 0.8, 1.0\}$ and for $R$ ranging over the divisors of 30 from smallest to largest. Fig 4a–4d show the performance for the GLRT algorithm assuming no observation noise ($\sigma_{\text{te}} = 0$), known $\gamma$ and known total driving variation $\sigma^2 = \sigma_{\text{co}}^2 + \sigma_{\text{bi}}^2 = 1$. Fig 4e–4h show the performance for the GLRT algorithm assuming known driving noise level $\sigma = 1$ and observational noise level $\sigma_{\text{te}} = 1$, while both $\gamma$ and $A$ are unknown to the algorithm.

The numerical simulations reflect the following observations implied by the analytical model.

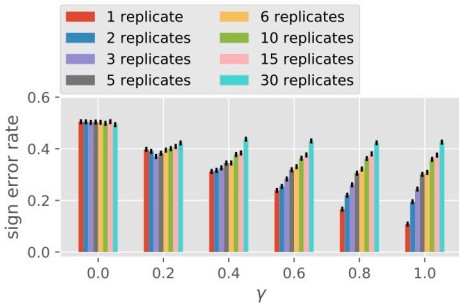

**(a)** One-shot sampling, known $\gamma$, $A = 0.1$, $\sigma_{\text{te}} = 0$.

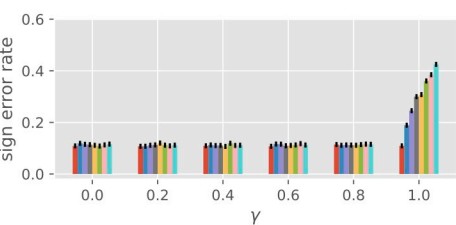

**(b)** Multi-shot sampling, known $\gamma$, $A = 0.1$, $\sigma_{\text{te}} = 0$.

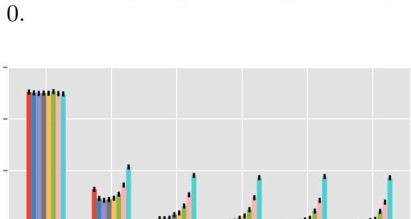

**(c)** One-shot sampling, known $\gamma$, $A = 0.5$, $\sigma_{\text{te}} = 0$.

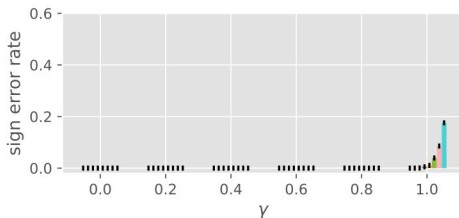

**(d)** Multi-shot sampling, known $\gamma$, $A = 0.5$, $\sigma_{\text{te}} = 0$.

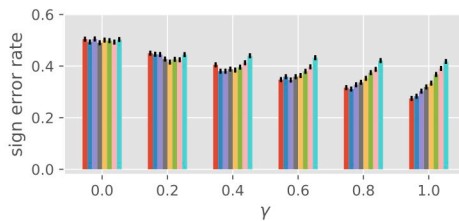

**(e)** One-shot sampling, unknown $\gamma$, $A = 0.1$, $\sigma_{\text{te}} = 1$.

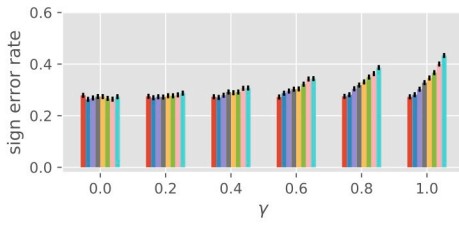

**(f)** Multi-shot sampling, unknown $\gamma$, $A = 0.1$, $\sigma_{\text{te}} = 1$.

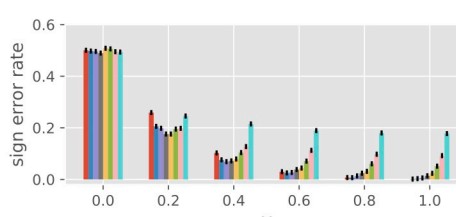

**(g)** One-shot sampling, unknown $\gamma$, $A = 0.5$, $\sigma_{\text{te}} = 1$.

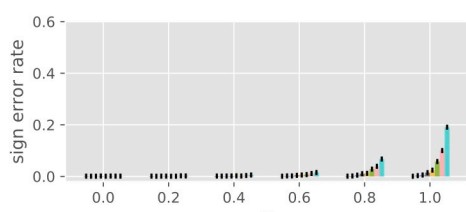

**(h)** Multi-shot sampling, unknown $\gamma$, $A = 0.5$, $\sigma_{\text{te}} = 1$.

**Fig 4. Performance of the GLRT in single-gene sign recovery with different numbers of replicates.**

1. Under one-shot sampling, when $\gamma = 0$, the GLRT is equivalent to random guessing.

2. The GLRT performs the same under one-shot and multi-shot sampling when $\gamma = 1$.

3. Under no observation noise, the performance for multi-shot sampling is the same for all $\gamma < 1$.

   Some empirical observations are in order.

1. Multi-shot sampling outperforms one-shot sampling (unless $\gamma = 1$, where they have the same error probability).

2. For one-shot sampling, the performance improves as $\gamma$ increases. Regarding the number of replicates $R$ per condition, if $\gamma = 0.2$ (small condition effect), a medium number of replicates (2 to 5) outperforms the single replicate strategy. For larger $\gamma$, one replicate per condition is the best.

3. For multi-shot sampling, performance worsens as $\gamma$ increases. One replicate per condition ($R = 1$) is best.

4. Comparing Fig 4a–4d vs. Fig 4e–4h, we observe that the performance degrades with the addition of observation noise, though for moderate noise ($\sigma_{te} = 1.0$) the effect of observation noise on the sign error is not large. Also, the effect of the algorithm not knowing $\gamma$ is not large.

**Simulations on multi-gene network reconstruction.** This subsection studies the case when multiple genes interact through the GRN. The goal is to compare one-shot vs. multi-shot sampling for BSLR under a variety of scenarios, including different homogeneous $\gamma$ values, varying number of replicates, varying observation noise level, and heterogeneous $\gamma$ values.

The performance evaluation for multi-gene network reconstruction is trickier than the single-gene case because of the many degrees of freedom introduced by the number of genes. First, the network adjacency matrix $A$ is now an $n$-by-$n$ matrix. While some notion of "size" of $A$ (like the spectral radius or the matrix norm) may be important, potentially every entry of $A$ may affect the reconstruction result. So instead of fixing a ground truth $A$ as in Fig 4, we fix a prior distribution of $A$ with split Gaussian prior described in S2 Appendix (note we assume the knowledge of no autoregulation), and choose $A$ i.i.d. from the prior distribution with $d_{max} = 3$. Second, because the prior of $A$ can be subject to sparsity constraints and thus far from a uniform distribution, multiple loss functions that are more meaningful than the ternary error rate can be considered for performance. So we consider ternary FDR, ternary FNR and ternary FPR for the multi-gene case. In the simulations we have 20 genes and $d_{max} = 3$ with in-degree uniformly distributed over $\{0, 1, \ldots, d_{max}\}$, so the average in-degree is 1.5. The number of sampling times is $T = 6$ and $CR = 30$.

**Varying $\gamma$, $R$ and $\sigma_{te}$.** In this set of simulations we fix the observation noise level and vary the number of replicates $R$ and the condition correlation coefficient $\gamma$. The performance of BSLR under one-shot and multi-shot sampling is shown in Fig 5 ($\sigma_{te} = 0$) and Fig 6 ($\sigma_{te} = 1$). Note BSLR does not apply to a single condition with 30 replicates due to the constraint that the degrees of freedom $C(T - 1) - k - 2$ in the second stage must be at least 1.

For one-shot sampling, when $\gamma = 0$, we see in both Figs 5 and 6 that BSLR is no different from random guessing, with an FDR close to $1 - \frac{1}{2}\frac{1.5}{19} \approx 0.96$ and an FNR and an FPR such that $l_{FNR} + \frac{1}{2}l_{FPR} \approx 1$ (recall the example of random guessing at the end of the section of the model for gene regulatory network topology). When $\gamma = 1$, BSLR performs similarly with one-shot or multi-shot sampling, which is consistent with property 1 in the section on general properties. As $\gamma$ increases from 0 to 1, under one-shot sampling for a fixed number of replicates, the FDR and FNR reduce greatly. For example, as $\gamma$ increases from 0.2 to 1, the FDR for single replicate under one-shot sampling decreases from 0.74 to 0.31 with noiseless data (Fig 5), and from 0.79 to 0.36 with noisy data (Fig 6), while the FNR decreases from 0.70 to 0.00 with noiseless data, and from 0.78 to 0.04 with noisy data. This decrease is more pronounced for smaller number of replicates. Note the trend of the performance of BSLR under one-shot sampling as a function of $R$ and $\gamma$ is very similar to that of GLRT in Fig 4e and 4g.

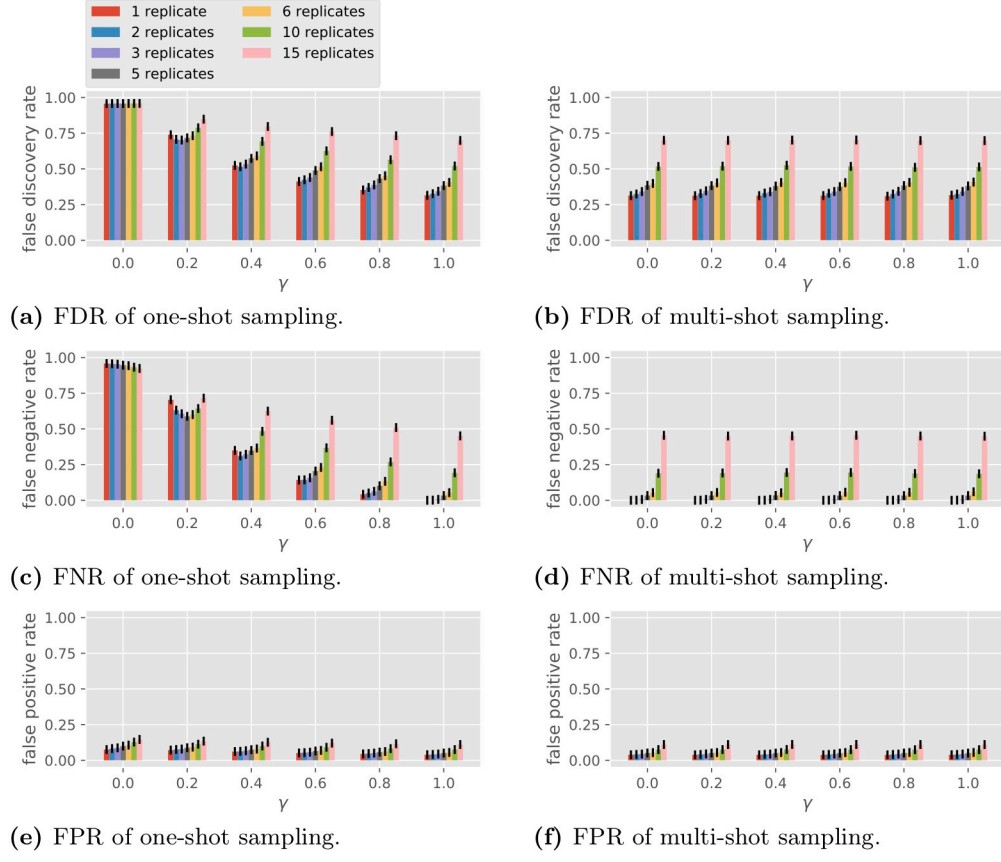

**Fig 5. Performance of the BSLR in multi-gene network reconstruction with different numbers of replicates, $\sigma_{\text{te}} = 0$.**

For multi-shot sampling, in the noiseless case, we see all three losses are invariant with respect to different $\gamma$ for fixed $R$, which is consistent with property 4 in the section on general properties because BSLR is an average-based scale-invariant algorithm (note $CR$ is a constant so for different $R$ the performance is different due to the change in $C$). In the noisy case, the FDR and FNR slightly decrease as $\gamma$ increases, which is an opposite trend compared with Fig 4f and 4h.

In summary, the main conclusions from Figs 5 and 6 are the following.

- The performance of BSLR under multi-shot sampling is consistently better than that under one-shot sampling.

- The performance of BSLR under one-shot sampling varies with $\gamma$, from random guessing performance at $\gamma = 0$ to the same performance as multi-shot sampling at $\gamma = 1$.

- By comparing Figs 5 with 6, we see the observation noise of $\sigma_{\text{te}} = 1$ has only a small effect on the performance with the two sampling methods.

**Reduced number of directly differentially expressed genes.** In the above simulations we have assumed all genes are equally directly differentially expressed. In other words, we took $\sigma_{\text{co},j}^2 + \sigma_{\text{bi},j}^2 = 1$ and $\sigma_{\text{co},j} = \sigma_{\text{co}}$ for all $j$. To test what happens more generally, we conducted simulations such that only half of the genes are directly differentially expressed genes (DDEGs),

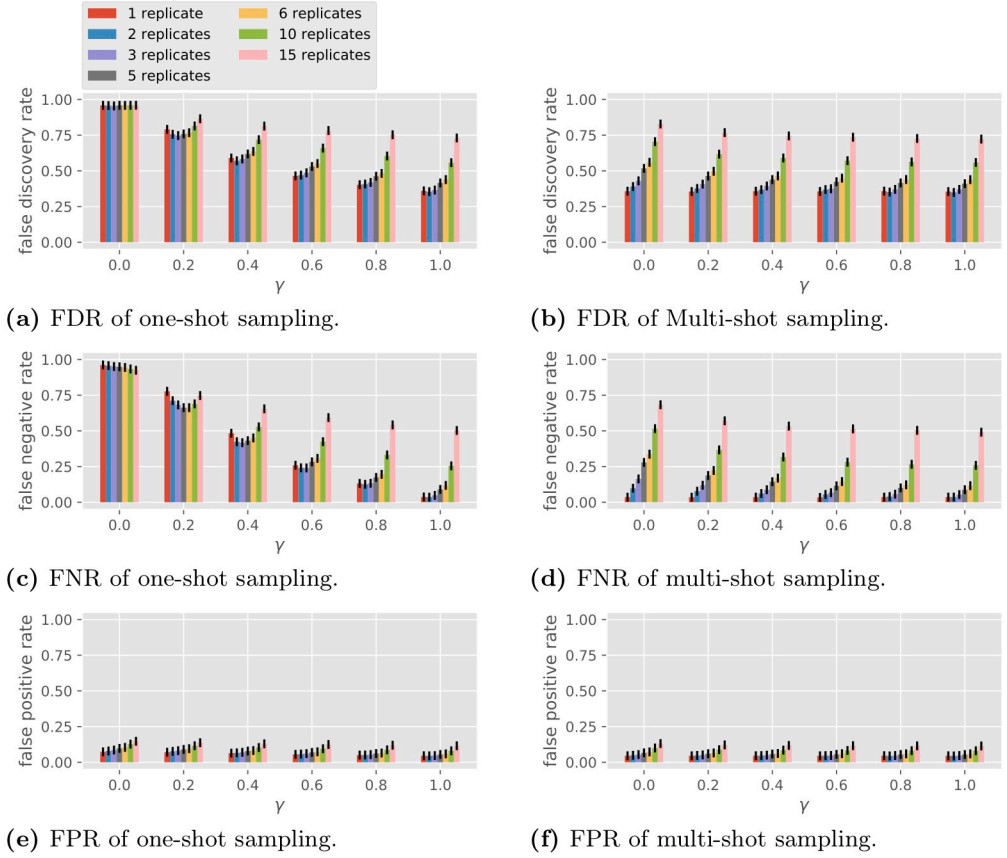

**Fig 6. Performance of the BSLR in multi-gene network reconstruction with different numbers of replicates, $\sigma_{\text{te}} = 1$.**

while the other half are non-DDEGs. To do so, we assign $\sigma^2_{\text{co},j} = 0.8$ and $\sigma^2_{\text{bi},j} = 0.2$ for $1 \leq j \leq 10$, and $\sigma^2_{\text{co},j} = 0$ and $\sigma^2_{\text{bi},j} = 1$ for $11 \leq j \leq 20$. The results for $R = 3$ are pictured in Fig 7. We see that with one-shot sampling the edges coming out of the DDEGs are reconstructed with lower FDR and FNR compared to those coming out of non-DDEGs. However, under one-shot sampling, even the edges from the non-DDEGs in Fig 7 are recovered with much lower FDR and FNR, as compared to one-shot sampling in Fig 6 with $\gamma = 0$ and $R = 3$ (both FDR and FNR are around 0.95). The results indicate that the performance of BSLR under one-shot sampling benefits from diversity in conditions even when not all genes are directly differentially expressed.

We summarize the simulations performed in Table 1. Note the last row is a summary of Table 2 in the Discussion section.

## Information limitation for reconstruction under one shot sampling without condition effect

In the previous section it is shown that both GLRT and BSLR are close to random guessing under one-shot sampling when $\sigma_{\text{co},j} = 0$ for all $j$. This leads us to the following question: is the network reconstruction with no condition effect ($\sigma_{\text{co},j} = 0$ for all $j$) information theoretically possible? In this section we examine this question under general estimator-independent

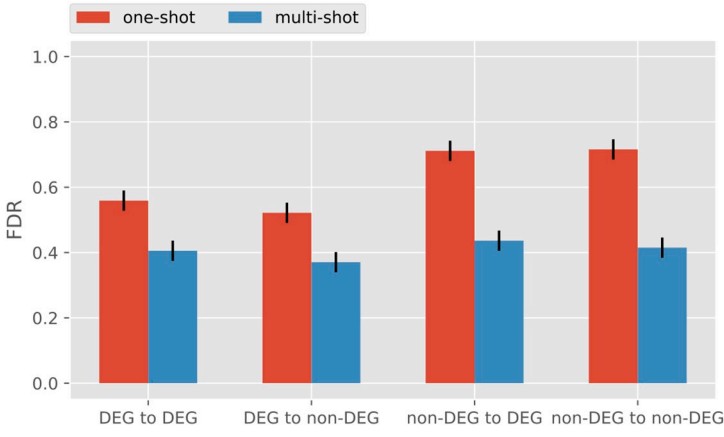

**(a)** FDR.

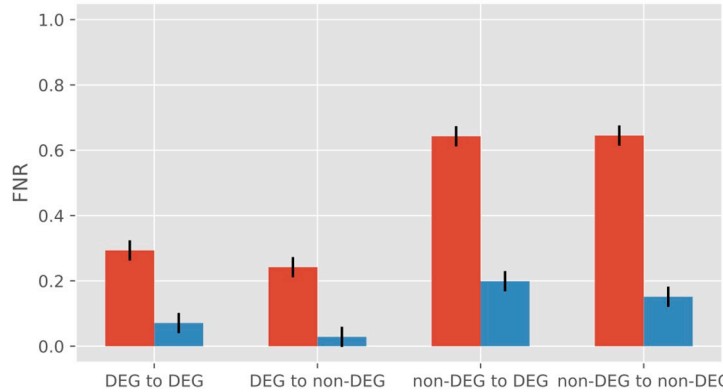

**(b)** FNR.

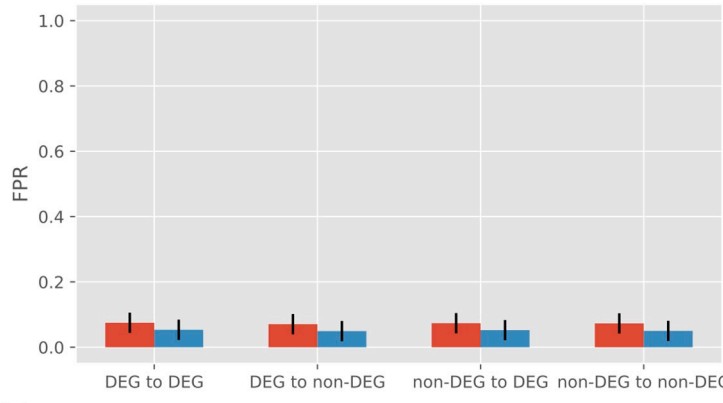

**(c)** FPR.

**Fig 7. Performance of BSLR for heterogeneous $\sigma_{\text{co},j}$ with $\sigma_{\text{te}} = 1$.**

settings. Note in this case the trajectories of all individuals are independent given $A$ regardless of $(c_k)_{k \in [K]}$.

As we have seen in Proposition 1 part 2, when $\Sigma_{\text{co}} = 0$, the distribution of the observed data $Y$ is invariant under adjacency matrix $A$ or $-A$, implying any estimator will have a sign error

**Table 1. Summary of simulation results.** OS and MS stand for the losses of one-shot sampling and multi-shot sampling. RG stands for random guessing. * indicates mathematically proved results.

| Result | Algorithm | Setting | Key observations |
|--------|-----------|---------|------------------|
| Fig 4 | GLRT | single-gene | • MS ≤ OS<br>• OS = RG at $\gamma = 0$*<br>• OS = MS at $\gamma = 1$*<br>• OS decreases with $\gamma$<br>• MS without noise is the same for $R = 1$ or $\gamma < 1$*<br>• MS with noise increases with $\gamma$<br>• optimal $R$ for OS depends on $\gamma$<br>• noise and unknown $\gamma$ slightly affects performance |
| Fig 5 | BSLR | multi-gene, noiseless | • MS ≤ OS<br>• OS = RG at $\gamma = 0$<br>• OS = MS at $\gamma = 1$*<br>• OS decreases with $\gamma$<br>• MS constant with $\gamma$ for given $R$* |
| Fig 6 | BSLR | multi-gene, noisy | • MS ≤ OS<br>• OS = RG at $\gamma = 0$<br>• OS = MS at $\gamma = 1$*<br>• OS decreases with $\gamma$<br>• noise slightly affects performance |
| Fig 7 | BSLR | multi-gene, heterogeneous $\gamma$ | • DDEG regulation is better recovered<br>• non-DDEG regulation is recovered better in the presence of DDEGs |
| Table 2 | BSLR | Locke model | • MS < OS with biologically plausible data<br>• OS is better with replicate averaging<br>• MS is better without replicate averaging |

probability no better than random guessing for the average or worst case over $A$ and $-A$. Here, instead of sign error probability, we consider the estimation for $A$ itself.

The extreme case with infinite number of samples available for network reconstruction is considered to give a lower bound on the accuracy for the finite data case. Note that with infinite number of samples a sufficient statistic for the estimation of the parameter $A$ is the marginal distributions of $X^1(t)$; no information on the correlation of $(X^1(t))_{t \in [T]}$ across time $t$ can be obtained. A similar observation is made in [7] for sampling stochastic differential equations.

We first consider the transient case with $X(0) = 0$ as stated in the section of the model for gene expression dynamics. With infinite data the covariance matrix $\Sigma_t \triangleq \mathrm{Cov}(X(t)) = \sum_{\tau=1}^{t} (A^*)^{t-\tau} A^{t-\tau}$ can be recovered for $t \in [T]$. Now we want to solve $A$ from $(\Sigma_t)_{t \in [T]}$. As a special case, if $A^*A = \rho I_n$ (i.e., $\rho^{-1/2} A$ is orthogonal) then we will have $\Sigma_t = \sum_{\tau=0}^{t-1} \rho^\tau I_n$. As a result, given $(\Sigma_t)_{t \in [T]}$ in the above form, no more information of $A$ can

**Table 2. BSLR evaluation on Locke network.** The errors are estimated using Hoeffding's inequality over 1000 simulations with significance level 0.05.

| Sampling method | Replicate averaging | FDR | FNR | FPR |
|-----------------|---------------------|-----|-----|-----|
| one-shot | yes | 0.71 ± 0.04 | 0.69 ± 0.04 | 0.54 ± 0.04 |
| one-shot | no | 0.74 ± 0.04 | 0.73 ± 0.04 | 0.54 ± 0.04 |
| multi-shot | yes | 0.67 ± 0.04 | 0.64 ± 0.04 | 0.55 ± 0.04 |
| multi-shot | no | 0.59 ± 0.04 | 0.55 ± 0.04 | 0.56 ± 0.04 |

be obtained other than $\rho^{-1/2} A$ being orthogonal, with $\frac{n(n-1)}{2}$ degrees of freedom remaining. In general case it is not clear if $A$ can be recovered from $(\Sigma_t)_{t \in [T]}$.

Now consider the case where $X^k$ is in steady state; i.e., $X(0)$ is random such that $\mathrm{Cov}(X(t))$ is invariant with $t$. With infinite amount of data we can get the covariance matrix $\Sigma$, which satisfies $\Sigma = A^* \Sigma A + I$. Since covariance matrices are symmetric, we have $\frac{n(n+1)}{2}$ equations for $n^2$ variables in $A$. Thus $A$ is in general not determined by the equation uniquely. In fact, note that $\Sigma$ is positive definite. Then by eigendecomposition $\Sigma = Q\Lambda Q^*$, where $Q$ is an orthogonal matrix and $\Lambda$ the diagonal matrix of the eigenvalues of $\Sigma$. Then $\Lambda = (Q^*AQ)^* \Lambda (Q^*AQ) + I$. Let $B = QAQ^*$. Then $\Lambda = B^* \Lambda B$. By the Gram–Schmidt process, $B$ can be determined with $\frac{n(n-1)}{2}$ degrees of freedom. So the network cannot be recovered from the stationary covariance matrix.

In summary, the recovery of the matrix $A$ is generally not possible in the stationary case, and also not possible in the transient case at least when $A$ is orthogonal. To reconstruct $A$, further constraints (like sparsity) may be required.

## Discussion

### One-shot sampling in the literature

This section reviews the sampling procedures reported in several papers measuring gene expression levels in biological organisms with samples collected at different times to form time series data. In all cases, the sampling is one-shot, in the sense that a single plant or cell is only sampled at one time.

**Microorganisms.** In the transcriptional network inference challenge from DREAM5 [8], three compendia of biological data sets were provided based on microorganisms (*E. coli*, *S. aureus*, and *S. cerevisiae*), and some of the data corresponded to different sampling times in a time series. Being based on microorganisms, the expression level measurements involved multiple individuals per sample, a form of one-shot sampling.

**Plants.** In [9], the plants are exposed to nitrate, which serves as a synchronizing event, and samples are taken from three to twenty minutes after the synchronizing event. The statement "… each replicate is independent of all microarrays preceding and following in time" suggests the experiments are based on one-shot sampling. In contrast, the state-space model with correlation between transcription factors in an earlier time and the regulated genes in a later time fits multi-shot sampling. [10] studied the gene expression difference between leaves at different developmental stages in rice. The 12th, 11th and 10th leaf blades were collected every 3 days for 15 days starting from the day of the emergence of the 12th leaves. While a single plant could provide multiple samples, namely three different leaves at a given sampling time, no plant was sampled at two different times. Thus, from the standpoint of producing time series data, the sampling in this paper was one-shot sampling. [11] devised the phenol-sodium dodecyl sulfate (SDS) method for isolating total RNA from *Arabidopsis*. It reports the relative level of mRNA of several genes for five time points ranging up to six hours after exposure to a synchronizing event, namely being sprayed by a hormone *trans*-zeatin. The samples were taken from the leaves of plants. It is not clear from the paper whether the samples were collected from different leaves of the same plant, or from leaves of different plants.

**Animals.** [12] likely used one-shot sampling for their −24, 60, 120, 168 hour time series data from mouse skeletal muscle C2C12 cells without specifying whether the samples are all taken from different individuals. [13] produced time series data by extracting cells from a human, seeding the cells on plates, and producing samples in triplicate, at a series of six times, for each of five conditions. Multiple cells are used for each sample with different sets of cells being used for different samples, so this is an example of one-shot sampling of *in vitro*

experiment in the sense that each plate of cells is one individual. The use of (five) multiple conditions can serve as a surrogate for a single individual set of cells to gain the effect of multi-shot sampling. Similarly, the data sets produced by [14] involving the plating of HeLa S3 cells can be classified as one-shot samples because different samples are made from different sets of individual cells. Interestingly, the samples are prepared under one set of conditions, so the use of different conditions is not adopted as a surrogate for multi-shot sampling. However, a particular line of cells was selected (HeLa S3) for which cells can be highly synchronized. Also, the paper does not attempt to determine causal interactions.

**In silico.** The three *in silico* benchmark suites described in the GeneNetWeaver paper on performance profiling of network inference methods [1] consisted of steady state, and therefore one-shot, samples from dynamical models. However, the GeneNetWeaver software can be used to generate multi-shot time series data, and some of that was included in the network inference challenges, DREAM3, DREAM4, and DREAM5 [1, 8].

## On biological replicates

In many biological experiments, independent biological replicates are used to reduce the variation in the measurements and to consequently increase the power of the statistical tests. It turns out that both how to use biological replicates, and the power of biological replicates, depend on whether the sampling is one-shot or multi-shot. To focus on this issue we first summarize how replicates have traditionally been used for the more common problem of gene differential expression analysis, before turning to the use of replicates for recovery of gene regulatory networks.

The following summarizes the use of replicates for gene differential expression analysis. A recent survey [15] suggests a minimum of three replicates for RNA-seq experiments whenever sample availability allows. Briggs et al. [16] studies the effect of biological replication together with dye switching in microarray experiments and recommends biological replication when precision in the measurements is desired. Liu et al. [17] studies the tradeoff between biological replication and sequencing depth under a sequencing budget limit in RNA-seq differential expression (DE) analysis. It proposes a metric for cost effectiveness that suggests a sequencing depth of 10 million reads per library of human breast cells and 2–6 biological replicates for optimal RNA-seq DE design. Schurch et al. [18] studies the number of necessary biological replicates in RNA-seq differential expression experiments on *S. cerevisiae* quantitatively with various statistical tools and concludes with the usage of a minimum of six biological replicates.

The choice of replication strategy depends on how the statistical algorithm uses the replicate data. In many differential analysis software packages replicates are treated as independent samples with identical experimental conditions. For example, in edgeR [19] and sleuth [20] the logarithm of the abundance of gene $i$ in sample $m$ is assumed to be $x_m^* \beta_i$, where $x_m$ is the column vector of design characteristics with respect to $p$ variates for sample $m$ and $\beta_i$ the column vector of the associated effects of the $p$ variates to gene $i$. Replicate samples can then be used to expand the design matrix $x$ with identical columns. Note that, as a result, replicates are not necessary for edgeR and sleuth because samples with different design characteristics can all contribute to the estimation of $\beta$. It is then not clear whether it is better to have more replicates under the same condition, or to have more conditions, for a fixed total number of samples.

For regulatory network reconstruction there is even less consensus on how replicates should be used. One straightforward way is to reduce the replicates into a single set of data by

averaging either directly or after a random resampling of the original replicated data. In this case the mean of the replicates are used as a better estimate of the population than each single replicate, while higher moments of the empirical distribution of the replicates are practically ignored. Another way adopted in [9] is to account for all four potential transitions between two replicates in two adjacent sampling times in their machine learning algorithm due to the one-shot nature of the replicates. In the next section, we illustrate why replicates should be used differently for one-shot and multi-shot sampling, in the context of recovering a circadian clock network model.

## A case study on *Arabidopsis* circadian clock network

As we have discussed earlier, the current expression datasets are prominently one-shot, making a direct comparison between one-shot and multi-shot sampling in real biological data difficult. The lack of a well-accepted ground truth of the gene regulatory network also makes performance evaluation hard, if not impossible. To test the applicability of the sampling models on real biological data, we generate expression data from a most-accepted *Arabidopsis* circadian clock model using stochastic differential equation (SDE) model similar to GeneNetWeaver with condition-dependent Brownian motions, and evaluate the performance of BSLR.

To extend the sampling models in this paper to the more biologically plausible SDE models, we model the individual and condition-dependent variations by independent and coupled Brownian motions. Following the *Arabidopsis* clock network in [21], we let genes 1, 2, 3, and 4 be *LHY*, *TOC1*, X and Y, and construct the following group of SDEs (the dark condition in [21] is assumed here).

$$\mathrm{d}x_{1,t}^k = \left( \frac{n_1 \left(z_{3,t}^k\right)^a}{g_1^a + \left(z_{3,t}^k\right)^a} - \frac{m_1 x_{1,t}^k}{k_1 + x_{1,t}^k} \right) \mathrm{d}t + \sigma_{\mathrm{co},1}^x x_{1,t}^k \mathrm{d}B_{\mathrm{co},1,t}^{x,c_k} + \sigma_{\mathrm{bi},1}^x x_{1,t}^k \mathrm{d}B_{\mathrm{bi},1,t}^{x,k},$$

$$\begin{aligned} \mathrm{d}y_{1,t}^k &= \left( p_1 x_{1,t}^k - r_1 y_{1,t}^k + r_2 z_{1,t}^k - \frac{m_2 y_{1,t}^k}{k_2 + y_{1,t}^k} \right) \mathrm{d}t \\ &\quad + \sigma_{\mathrm{co},1}^y y_{1,t}^k \mathrm{d}B_{\mathrm{co},1,t}^{y,c_k} + \sigma_{\mathrm{bi},1}^y y_{1,t}^k \mathrm{d}B_{\mathrm{bi},1,t}^{y,k}, \end{aligned}$$

$$\mathrm{d}z_{1,t}^k = \left( r_1 y_{1,t}^k - r_2 z_{1,t}^k - \frac{m_3 z_{1,t}^k}{k_3 + z_{1,t}^k} \right) \mathrm{d}t + \sigma_{\mathrm{co},1}^z z_{1,t}^k \mathrm{d}B_{\mathrm{co},1,t}^{z,c_k} + \sigma_{\mathrm{bi},1}^z z_{1,t}^k \mathrm{d}B_{\mathrm{bi},1,t}^{z,k},$$

$$\begin{aligned} \mathrm{d}x_{2,t}^k &= \left( \frac{n_2 \left(z_{4,t}^k\right)^b}{g_2^b + \left(z_{4,t}^k\right)^b} \frac{g_3^c}{g_3^c + \left(z_{1,t}^k\right)^c} - \frac{m_4 x_{2,t}^k}{k_4 + x_{2,t}^k} \right) \mathrm{d}t \\ &\quad + \sigma_{\mathrm{co},2}^x x_{2,t}^k \mathrm{d}B_{\mathrm{co},2,t}^{x,c_k} + \sigma_{\mathrm{bi},2}^x x_{2,t}^k \mathrm{d}B_{\mathrm{bi},2,t}^{x,k}, \end{aligned}$$

$$\begin{aligned} \mathrm{d}y_{2,t}^k &= \left( p_2 x_{2,t}^k - r_3 y_{2,t}^k + r_4 z_{2,t}^k - (m_5 + m_6) \frac{y_{2,t}^k}{k_5 + y_{2,t}^k} \right) \mathrm{d}t \\ &\quad + \sigma_{\mathrm{co},2}^y y_{2,t}^k \mathrm{d}B_{\mathrm{co},2,t}^{y,c_k} + \sigma_{\mathrm{bi},2}^y y_{2,t}^k \mathrm{d}B_{\mathrm{bi},2,t}^{y,k}, \end{aligned}$$

$$dz_{2,t}^k = \left( r_3 y_{2,t}^k - r_4 z_{2,t}^k - (m_7 + m_8)\frac{z_{2,t}^k}{k_6 + z_{2,t}^k} \right) dt + \sigma_{co,2}^z z_{2,t}^k dB_{co,2,t}^{z,c_k} + \sigma_{bi,2}^z z_{2,t}^k dB_{bi,2,t}^{z,k},$$

$$dx_{3,t}^k = \left( \frac{n_3 (z_{1,t}^k)^d}{g_4^d + (z_{1,t}^k)^d} - \frac{m_9 x_{3,t}^k}{k_7 + x_{3,t}^k} \right) dt + \sigma_{co,3}^x x_{3,t}^k dB_{co,3,t}^{x,c_k} + \sigma_{bi,3}^x x_{3,t}^k dB_{bi,3,t}^{x,k},$$

$$dy_{3,t}^k = \left( p_3 x_{3,t}^k - r_5 y_{3,t}^k + r_6 z_{3,t}^k - \frac{m_{10} y_{3,t}^k}{k_8 + y_{3,t}^k} \right) dt$$
$$+ \sigma_{co,3}^y y_{3,t}^k dB_{co,3,t}^{y,c_k} + \sigma_{bi,3}^y y_{3,t}^k dB_{bi,3,t}^{y,k},$$

$$dz_{3,t}^k = \left( r_5 y_{3,t}^k - r_6 z_{3,t}^k - \frac{m_{11} z_{3,t}^k}{k_9 + z_{3,t}^k} \right) dt + \sigma_{co,3}^z z_{3,t}^k dB_{co,3,t}^{z,c_k} + \sigma_{bi,3}^z z_{3,t}^k dB_{bi,3,t}^{z,k},$$

$$dx_{4,t}^k = \left( \frac{n_5 g_5^e}{g_5^e + (z_{2,t}^k)^e}\frac{g_6^f}{g_6^f + (z_{1,t}^k)^f} - \frac{m_{12} x_{4,t}^k}{k_{10} + x_{4,t}^k} \right) dt$$
$$+ \sigma_{co,4}^x x_{4,t}^k dB_{co,4,t}^{x,c_k} + \sigma_{bi,4}^x x_{4,t}^k dB_{bi,4,t}^{x,k},$$

$$dy_{4,t}^k = \left( p_4 x_{4,t}^k - r_7 y_{4,t}^k + r_8 z_{4,t}^k - \frac{m_{13} y_{4,t}^k}{k_{11} + y_{4,t}^k} \right) dt$$
$$+ \sigma_{co,4}^y y_{4,t}^k dB_{co,4,t}^{y,c_k} + \sigma_{bi,4}^y y_{4,t}^k dB_{bi,4,t}^{y,k},$$

$$dz_{4,t}^k = \left( r_7 y_{4,t}^k - r_8 z_{4,t}^k - \frac{m_{14} z_{4,t}^k}{k_{12} + z_{4,t}^k} \right) dt + \sigma_{co,4}^z z_{4,t}^k dB_{co,4,t}^{z,c_k} + \sigma_{bi,4}^z z_{4,t}^k dB_{bi,4,t}^{z,k}.$$

Here $x_{i,t}^k$, $y_{i,t}^k$, and $z_{i,t}^k$ denote the mRNA abundance, and cytoplasmic and nuclear protein concentrations of gene $i$ at time $t$ in plant $k$. The $B$ terms are independent standard Brownian motions (Wiener processes). Note the linear diffusion terms attenuate the Brownian motions as the processes get close to 0 and consequently keep the processes nonnegative. Compared to the GLM analyzed in this paper, this SDE model captures the nonlinearity of the regulatory interactions, the continuous-time nature of the dynamics, and the detailed diffusion from mRNA to cytoplasmic and nuclear protein. So the SDE model is much more complicated and considered a basic version of the state-of-the-art circadian clock model (see [22, 23]). Nevertheless, the SDE model shares a property with the GLM that allows the gene regulatory interactions to be summarized by a single signed directed graph: the effect of increasing the mRNA abundance of one gene on that of another has a constant sign regardless of the mRNA abundances and the protein concentrations of any genes. For example, the drift of $x_1$ (mRNA of *LHY*) would have a tendency of increasing with an increased $x_3$ (mRNA of X) through the equations for $y_3$ (cytoplasmic protein of X) and $z_3$ (nuclear protein of X) regardless of the specific values of all the $x$'s, $y$'s and $z$'s. Fig 8 shows the signed directed graph for the SDE model of Locke network. Now we have a ground-truth network based on the SDE model.

We choose the parameters in the drift coefficients (the terms in front of the d$t$'s) accordingly to the supplementary material of [21], where the authors optimized the parameters to fit experimental results. We sample the SDE at times 0, 2, 4, 6, 8, and 10 for a single condition ($C = 1$) with three replicates ($R = 3$), $\sigma_{co} = 0.3$ and $\sigma_{bi} = 0.4$, and obtain the performance of BSLR in Table 2. The estimates and the errors are based on 1000 simulations for each sampling

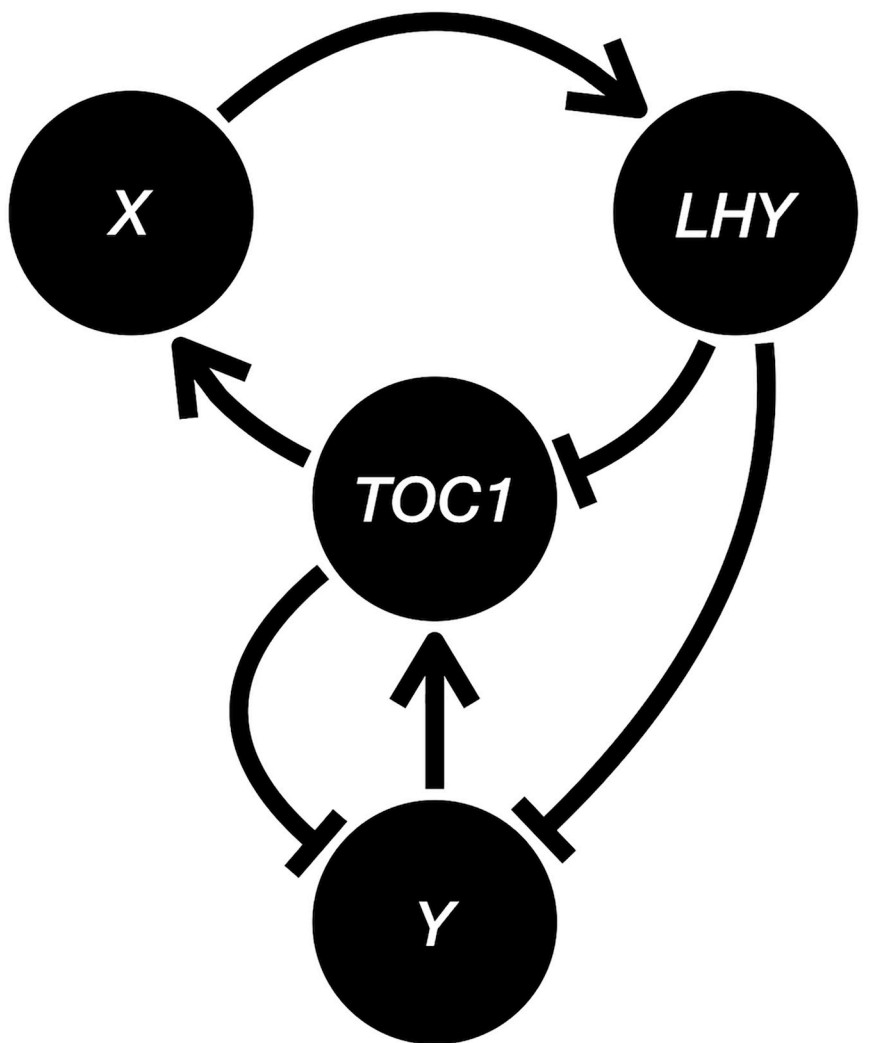

**Fig 8. The signed directed graph for the Locke network.**

method. Here the significance level of the Granger causality test in BSLR is set to 0.5. Note a random guess would have FDR = 0.75 and FNR $+ \frac{1}{2}$FPR $= 1$. We see from Table 2 that BSLR without replicate averaging on one-shot sampling data is no different from random guessing, while BSLR with replicate averaging doing slightly better in FDR and FNR. This is because replicate averaging of one-shot data practically increases the condition effect by reducing the biological variation, and thus gets a better temporal correlation between one-shot samples of adjacent times. BSLR performs better on multi-shot data compared to one-shot data because the biological variations of the previous times also contribute to the temporal correlation. In particular, BSLR without replicate averaging on the multi-shot data has the best performance because it allows tracking the individual replicates rather than merely tracking their averages. Although the performance numbers appear far from ideal, this demonstrates remarkable improvement from BSLR with replicate averaging on one-shot data to BSLR without replicate averaging on multi-shot data, especially considering the highly nonlinear SDE model, the unobserved protein concentrations levels, the very limited number of 18 samples (3 replicates

with 6 times) and the fact that BSLR does not use any knowledge of the (around 60) parameters or the form of the equations, highlighting the difference in the statistical power of one-shot and multi-shot data and its implication in downstream statistical analysis decisions (replicate averaging vs. no replicate averaging).

In summary, we demonstrated a setting of the biologically plausible *Arabidopsis* circadian clock network with a single condition, where the BSLR performs similarly to a random guessing algorithm under one-shot sampling, and performs significantly better under multi-shot sampling. We also show that whether replicate averaging should be done or not varies with the sampling method.

## Conclusions

One-shot sampling can miss a lot of potentially useful correlation information. Often gene expression data collected from plants is prepared under one-shot sampling. One factor that can partially mitigate the shortcomings of one-shot sampling is to prepare samples under a variety of conditions or perturbations. One-shot samples grown under the same condition can then be thought of as a surrogate for the multi-shot samples of an individual plant.

To clarify issues and take a step towards quantifying them, we proposed a gene expression dynamic model for gene regulatory network reconstruction that explicitly captures the condition variation effect. We show analytically and numerically the performance of two algorithms for single-gene and multi-gene settings. We also demonstrate the difficulty of network reconstruction without condition variation effect.

There is little agreement across the biology literature about how to model the impact of condition on the gene regulatory network. In some cases, it is not even clear that we are observing the same network structure as conditions vary. Nevertheless, our results suggest that the preparation of samples under different conditions can partially compensate for the shortcomings of one-shot sampling.

## Supporting information

**S1 Appendix. Joint estimation for single-gene autoregulation recovery.** The parameters $A$, $\gamma$, $\sigma$, and $\sigma_{te}$ are assumed unknown and jointly estimated in GLRT.
(PDF)

**S2 Appendix. Split Gaussian network prior.** The random network prior distribution used to generate the multi-gene network.
(PDF)

## Author Contributions

**Writing – original draft:** Xiaohan Kang, Bruce Hajek.

**Writing – review & editing:** Xiaohan Kang, Bruce Hajek, Faqiang Wu, Yoshie Hanzawa.

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
