## [Decision Letter · Decision Letter 0]

21 Aug 2019

PONE-D-19-16969

Time series experimental design under one-shot sampling: The importance of condition diversity

PLOS ONE

Dear Dr. Kang,

Thank you for submitting your manuscript to PLOS ONE. After careful consideration, we feel that it has merit but does not fully meet PLOS ONE’s publication criteria as it currently stands. Therefore, we invite you to submit a revised version of the manuscript that addresses the points raised during the review process.

In the reviewer comments below, I would emphasize the exploration of actual biological data and the comparison of performance to one or two other methods.

We would appreciate receiving your revised manuscript by Oct 05 2019 11:59PM. To enhance the reproducibility of your results, we recommend that if applicable you deposit your laboratory protocols in protocols.io, where a protocol can be assigned its own identifier (DOI) such that it can be cited independently in the future. For instructions see: http://journals.plos.org/plosone/s/submission-guidelines#loc-laboratory-protocols

We look forward to receiving your revised manuscript.

Kind regards,

Steven M. Abel, Ph.D.

Academic Editor

PLOS ONE

Journal Requirements:

Reviewers' comments:

Reviewer's Responses to Questions

**Comments to the Author**

1. Is the manuscript technically sound, and do the data support the conclusions?

Reviewer #1: Partly

2. Has the statistical analysis been performed appropriately and rigorously? 

Reviewer #1: Yes

3. Have the authors made all data underlying the findings in their manuscript fully available?

Reviewer #1: Yes

4. Is the manuscript presented in an intelligible fashion and written in standard English?

Reviewer #1: Yes

5. Review Comments to the Author

Reviewer #1: This papers explores the effect of condition variance and biological variance under one-shot and multi-shot sampling. It provides a general model for network representation when these parameters are known. Further, the authors describe the relationship between the sampling regimes and their representation in the model described. Finally, the authors simulation from this model and describe the performance of two-estimators under different configurations.

Things I would like to see:

- A table describing the simulations performed and summarizing the behavior

- A description of how the adjacency matrix A was chosen from simulations

- An application on "real" data

- A comparison of performance to other methods on some of the DREAM data

The section "On biological replicates" is a bit odd and seems tacked on. Particularly, the discussion of differential expression tools seems odd. The goal in those tools is not to infer gene regulation, but to infer whether the expression of some set of genes is changed given some experimental perturbation. Not sure what you mean here (line 551):

> It is then not clear whether replicates bring more benefit than the sheer additional amount of data compared to samples under different conditions.

6. PLOS authors have the option to publish the peer review history of their article (what does this mean?). If published, this will include your full peer review and any attached files.

Reviewer #1: No

---

## [Editor Report · Decision Letter 1]

17 Oct 2019

Time series experimental design under one-shot sampling: The importance of condition diversity

PONE-D-19-16969R1

Dear Dr. Kang,

We are pleased to inform you that your manuscript has been judged scientifically suitable for publication and will be formally accepted for publication once it complies with all outstanding technical requirements.

With kind regards,

Steven M. Abel, Ph.D.

Academic Editor

PLOS ONE

---

## [Editor Report · Acceptance letter]

22 Oct 2019

PONE-D-19-16969R1 

Time series experimental design under one-shot sampling: The importance of condition diversity 

Dear Dr. Kang:

I am pleased to inform you that your manuscript has been deemed suitable for publication in PLOS ONE. Congratulations! Your manuscript is now with our production department. 

With kind regards,

on behalf of

Dr. Steven M. Abel 

Academic Editor

PLOS ONE